# Reference Gene Selection for Expression Analyses by qRT-PCR in *Dendroctonus valens*

**DOI:** 10.3390/insects11060328

**Published:** 2020-05-27

**Authors:** Chunchun Zheng, Dongfang Zhao, Yabei Xu, Fengming Shi, Shixiang Zong, Jing Tao

**Affiliations:** Key Laboratory of Beijing for the Control of Forest Pests, Beijing Forestry University, Beijing 100083, China; zcc0426@163.com (C.Z.); zhaodf62@163.com (D.Z.); xuyabei@bjfu.edu.cn (Y.X.); shifengming@bjfu.edu.cn (F.S.); zongshixiang@bjfu.edu.cn (S.Z.)

**Keywords:** *Dendroctonus valens*, qRT-PCR, reference genes

## Abstract

*Dendroctonus valens* is the main pest of the genus *Pinus*. To facilitate gene expression analyses, suitable reference genes for adults and mature larvae of *D. valens* under different temperature conditions were determined. In particular, we obtained the sequences of candidate reference genes, *ACT*, *TUB*, *SHDA*, *PRS18*, *18S rRNA*, and *CYP4G55*, from transcriptome data. Real-time quantitative PCR was used to analyze gene expression, and geNorm, NormFinder, and BestKeeper were used to evaluate expression stability. Under different temperature conditions, the expression levels of *18S rRNA*, *PRS18*, and *TUB* were stable in adults, in which *18S rRNA* > *PRS18* > *TUB*. In mature larvae, the expression levels of *TUB*, *18S rRNA*, and *SDHA* were stable, in which *TUB* > *18S rRNA* > *SDHA*. The combination of *18S rRNA* and *PRS18* is recommended for studies of gene expression in adults and the combination of *18S rRNA* and *TUB* is effective for studies of gene expression in mature larvae of *D. valens* under different temperature conditions.

## 1. Introduction

*Dendroctonus valens* (Coleoptera: Scolytidae) is a major pest of the genus *Pinus*, native to Canada and North America. In China, abundant food resources, the lack of interspecific competition, and the lack of effective natural enemies have enabled the spread of *D. valens* [1]. Since its discovery in Shanxi Province in the late 1990s, the area of infestation has expanded rapidly, reaching Beijing, Hebei, Henan, Inner Mongolia, and Liaoning. The pest has caused serious harm to *Pinus tabulaeformis* and *P. Sylvestris* and has become one of the main invasive pests in China.

Precise analyses of gene expression levels are important for molecular biology research. Real-time quantitative PCR (qRT-PCR) has become the “gold standard” for transcript-level expression analyses, owing to its high accuracy, specificity, sensitivity, and rapidity [2,3]. However, in the implementation of qRT-PCR, RNA extraction, polymerase amplification, and cDNA synthesis can all lead to systematic errors [2,4]. To eliminate these sources of error, genes with constant expression levels across conditions are usually selected as references for normalization. Under different experimental treatments, the expression level of an ideal reference gene remains constant [5]. However, the wide application of qRT-PCR has revealed that a single gene is not expressed stably under complex experimental treatments [6,7]. Appropriate reference genes may differ depending on the insect species, experimental treatment, development stage, tissue type, and temperature [8,9,10,11,12]. Common reference genes for insect studies include beta-actin, beta-tubulin, alpha-tubulin, ribosomal protein S18 (*PRS18*), 18S ribosomal RNA (*18S rRNA*), and succinate dehydrogenase complex subunit A (*SDHA*) [13]. These genes are involved in normal metabolic processes in cells. Reference gene screening has been performed in Coleoptera, including *Leptinotarsa decemlineata*, *Diabrotica virgifera virgifera*, *Colaphellus bowringi,* and *Harmonia axyridis* [14,15,16,17].

Few studies have evaluated *D. valens* at the molecular level. Cano-Ramirez studied *P450* expression in the antennae and intestines of *D. valens* exposed to monoterpenes; the cytochrome *P450* gene *CYP4G55* was stably expressed under these conditions and therefore was identified as a suitable reference gene [18]. However, it is not clear whether this reference gene exhibits stable expression under other experimental treatments. In this study, we further screened reference genes to obtain a reliable internal reference gene for studying the gene expression pattern of *D. valens*.

## 2. Materials and Methods

### 2.1. Insects

Mature larvae and adults of *D. valens* were collected from Chifeng (Inner Mongolia, China). Live insects were brought to the laboratory in dark conditions. All samples were maintained for 2 h at 25 °C; some were placed in liquid nitrogen for quick-freezing and the rest were maintained at −10 °C, −5 °C, 0 °C, 5 °C and 10 °C for 1 h, immediately frozen in liquid nitrogen, and stored in a refrigerator at −80 °C.

### 2.2. RNA Isolation and cDNA Synthesis

TRIzol (No. 15596018; Invitrogen, Carlsbad, CA, USA) and the RNeasy Plus Mini Kit (No. 74134; Qiagen, Hilden, Germany) were used for RNA extraction from mature larvae and adults of *D. valens* according to the manufacturers’ instructions. The micro-ultraviolet/visible spectrophotometer NanoDrop 8000 (Thermo, Waltham, MA, USA) was used to determine the A260/A280 ratio for evaluations of the quality and concentration of extracted RNA. Single stranded cDNA was synthesized from 1.0 μg total RNA using the Prime Script RT Reagent Kit with gDNA Eraser Kit (TaKaRa, Shiga, Japan).

### 2.3. Selection of Candidate Reference Genes

Based on transcriptome data obtained by high-throughput sequencing in our laboratory, gene sequences were compared with sequences of related species in GenBank. Finally, six common housekeeping genes were selected as candidate reference genes (*ACT*, *TUB*, *SHDA*, *PRS18*, *18S rRNA*, and *CYP4G55*). The sequences of the candidate reference genes were verified by RT-PCR. Total RNA was extracted and cDNA was obtained by reverse transcription as described above. Primers for RT-PCR were designed, and cDNA was used as the template for PCR amplification. The PCR system (25 μL) included cDNA template (1 μL), 2× PrimeSTAR Max Premix (12.5 μL), forward and reverse primers (10 μmol/L; 0.5 μL), and ddH_2_O (10.5 μL). PCR conditions were as follows: 94 °C for 2 min; 35 cycles of 94 °C for 30 s, 58 °C for 30 s, 72 °C for 30 s; 72 °C for 3 min.

### 2.4. Primer Design

Based on the nucleotide sequences of candidate reference genes, Primer3Plus was used to design primers (Appendix A) and the primer sequences were synthesized by Beijing Ruiboxingke Biotechnology Co., Ltd. (Beijing, China). The specificity of the primers was determined by the confirmation of a single peak in the melting curve. A 5× gradient dilution of the cDNA of adults and mature larvae was used as a template to draw the standard curve to determine the amplification efficiency of primers.

### 2.5. qRT-PCR Analysis

cDNA was used as a template for qRT-PCR using the newly designed primers. The analysis was repeated three times for each sample with three technical repetitions for each group. The reaction system (12.5 μL) was as follows: cDNA template (1 μL), forward and reverse primers (10 μmol/L; 2 μL), SYBRPremix Ex Taq II (6.25 μL), and ddH_2_O (4.25 μL). The reaction conditions were as follows: 95 °C for 3 min; 40 cycles of 95 °C for 10 s, 58 °C for 30 s; 5 s at 95 °C, 65 °C–95 °C, 0.5 °C increase/cycle. After the reaction, the amplification curve and melting curve were confirmed.

### 2.6. Analysis and Verification of the Stability of Reference Genes

The experimental data obtained by qRT-PCR were analyzed by four methods: ΔCt, geNorm, BestKeeper, and NormFinder. The standard procedures for each method were followed. Based on the expression level of each candidate reference gene and results for each algorithm, stable expression across different temperatures was evaluated.

For verification, *HSP21*, which encodes a heat shock protein in *D. valens,* was used as a target gene. With different candidate reference genes, the 2^−ΔΔCt^ method was used to calculate the relative expression of *HSP21* under different temperature conditions [19] to verify the stability of reference genes.

## 3. Results

### 3.1. Selection of Candidate Reference Genes and Primer Design

*ACT*, *SDHA*, *18S rRNA*, *RPS18*, *CYP4G55*, and *TUB* were selected as candidate reference genes based on transcriptome sequencing data. An alignment showed high sequence similarity with *Dendroctonus ponderosae* loci of over 90%, and the sequences determined by RT-PCR were similar to those obtained by transcriptome sequencing (over 99% similarity). Specific primers for qPCR were designed based on gene sequences. A standard curve analysis indicated that the amplification efficiency of the qPCR primers was 91.3–109.8% with high correlation coefficients (*R*^2^ > 99%), indicating that the standard curve showed a good linear relationship (Table 1). In addition, the melting curves for all candidate genes exhibited single peaks, indicating high primer specificity (Appendix A).

### 3.2. Stability Analysis of Candidate Reference Genes

#### 3.2.1. Expression Levels of the Candidate Reference Genes

Transcript abundance and cycle threshold (Ct) variation are important parameters for screening reference genes. The stability of Ct values is an important criterion for reference gene selection. The smaller the Ct value of the gene, the higher its expression level; conversely, the larger the Ct value, the lower its expression level. As shown in Figure 1, the average Ct values for the six candidate reference genes were between 19 and 35 at different temperatures, and transcript abundances in adults were higher than those in mature larvae. The expression patterns of the six candidate reference genes were similar at the two developmental stages. The transcript abundances of *TUB* and *CYP4G55* were higher than those of other genes, while the abundance of *ACT* was the lowest. In addition, the average standard deviation of each gene set negatively correlated with its stability. Ct values for *CYP4G55* and *TUB* exhibited high variation among adults, while Ct values for *CYP4G55* and *ACT* exhibited substantial variation among mature larvae.

#### 3.2.2. geNorm Analysis

geNorm was used to analyze the expression stability of the candidate reference genes at different temperatures. Stability values (M) were obtained. A lower value indicates more stable gene expression. As shown in Figure 2, *TUB* and *18S rRNA* were the most stable and *CYP4G55* was the most unstable gene in both adults and mature larvae exposed to different temperatures. In Figure 3, we found that the pairwise variation Vn/(Vn + 1) was greater than 0.15 for samples at different temperatures. Therefore, to reduce experimental error, it is necessary to use multiple reference genes to analyze target gene expression in *D. valens* under different temperature treatments.

#### 3.2.3. NormFinder Analysis

The results of a NormFinder analysis differed slightly from those of the geNorm analysis, as shown in Figure 4. Using NormFinder, under the same experimental conditions, the most stable reference gene in adults was *18S rRNA* with a stability value of 0.26, followed by *PRS18* and *TUB* with stability values of 0.31 and 0.37. In mature larvae, *TUB* was the most stable gene (M = 0.25), followed by *SDHA* and *18S rRNA*, with stability values of 0.28 and 0.45. However, the most unstable reference gene for both adults and mature larvae was *CYP4G55*.

#### 3.2.4. BestKeeper Analysis

In a BestKeeper analysis, a higher correlation coefficient (*r*) indicates a lower standard deviation (SD) and coefficient of variation (CV) and higher reference gene stability. In this analysis, *SDHA* and *PRS18* were the most stable genes in adults, and *TUB* and *18S rRNA* were the most stable genes in mature larvae (Table 2 and Table 3). Moreover, the *p*-values for the candidate reference genes were all less than 0.05, indicating that these genes could be used in combination as co-reference genes.

### 3.3. Comprehensive Ranking of Candidate Reference Genes

According to the results obtained using the three algorithms, reference gene sequences were obtained. The geometric mean parameter values for each reference gene sequence were calculated as the final ranking [20]. As shown in Table 4, under different temperature treatments, the order of reference genes ranked by stability in adults was *18S rRNA* > *PRS18* > *TUB* > *SDHA* > *ACT* > *CYP4G55* and in mature larvae was *TUB* > *18S rRNA* > *SDHA* > *PRS18* > *CYP4G55* > *ACT*.

### 3.4. Verification of Reference Genes

To verify the stability of the reference genes, *18S rRNA* (the gene with the highest stability in comprehensive analyses in adults), *PRS18* (with slightly less stability), and *CYP4G55* (with the lowest stability) were selected to determine the relative expression level of *HSP21* in adults under different low-temperature conditions. As shown in Figure 5, using *18S rRNA* and *PRS18* as reference genes, levels of gene expression were similar. When *18S rRNA* was used as the reference gene, the expression level of *HSP21* under low temperatures was significantly lower than that in the 25 °C. When *PRS18* was used as the reference gene, the expression level of *HSP21* under temperature stress was slightly different from that of *18S rRNA* but was also lower than that in the 25 °C. However, when *CYP4G55* was used as the reference gene, the expression levels of the target gene *HSP21* at 10 °C and 5 °C were not significantly lower than those in the 25 °C, in contrast to the results obtained using *18S rRNA* or *PRS18* as the reference gene. This indicates that an unstable reference gene may affect the accuracy of qPCR results and even lead to incorrect conclusions.

## 4. Discussion

Our results showed that *18S rRNA* and *TUB* were stable reference genes in adults and mature larvae at different temperatures, respectively. *18S rRNA* is a commonly used reference gene. In *Coccinella septempunctata*, *18S rRNA* is considered the best reference gene for analyses of gene expression in different tissue types; in *H. axyridis*, it is considered the best reference gene for analyses of gene expression at different temperatures [17,21]. Tubulin is a cytoskeleton component. It functions in the maintenance cell shape, mitosis, cell movement, intracellular transport, and organelle composition. Genes encoding tubulin are often used as reference genes. *TUB* is expressed stably in various tissues of *D. virgifera virgifera* [15]. It is also stably expressed under different temperature treatments in *Nilaparvata lugens* [7], but is not stable in second-instar larvae of *Galeruca daurica* under different temperature treatments [22].

Few studies have focused on reference genes in *D. valens*. Cano-Ramirez preliminarily screened reference genes and found that *CYP4G55* is stable and could be used as a reference for studies of *P450* in the antennae and intestines of *D. valens* exposed to monoterpenes [18]. However, in this study, *CYP4G55* stability was low relative to those of the other candidates in both adults and mature larvae in low temperatures. Actin (ACT) is the main structural protein of the cytoskeleton with an important role in cell functions. However, we found that this gene, which is commonly used as a reference in other insects, is far less stable than other candidate reference genes in adults and mature larvae. Similarly, in different tissues of *C. septempunctata* and at different temperatures in *Phenacoccus solenopsis*, *Henosepilachna vigintioctomaculata*, and *Hippodamia convergens*, *ACT* has low stability [21,23,24,25]. It is a relatively stable reference gene in *Myzus persicae* under temperature stress [26]. These results prove that reference gene screening results depend on the species, sample type, and experimental conditions, further supporting the importance of screening.

Research has shown that none of the genes exhibited constant expression levels. Therefore, the use of two or more reference genes may yield more accurate results. Under different temperatures, *GAPDH* and *EF-1α* are stably expressed in *Spodoptera litura* [27], and *Actin*, *Mnf,* and *α-TUB* are stably expressed in *Drosophila melanogaster* [28]. In addition, for the same insect taxa and experimental conditions, different candidate genes can be identified. For example, a study has shown that *RPS15* and *RPL27* are stably expressed in *Helicoverpa armigera* [29], while another study has shown that the combination of *RPL28* and *RPS15* is the most suitable reference under the same conditions [30].

In this study, geNorm, NormFinder, and BestKeeper were used to evaluate and verify the stability of reference genes. The results obtained using the three algorithms were not completely consistent. Under different temperatures, geNorm, NormFinder, and BestKeeper all showed that *TUB* is the most stable reference genes in mature larvae. However, in adults exposed to different temperatures, geNorm and NormFinder both showed that *18S rRNA* and *PRS18* were relatively stable reference genes, while BestKeeper showed that *SDHA* and *PRS18* were relatively stable. Similar differences have been reported in studies of *Lipaphis erysimi* and *Anastrepha obliqua*; differences in software and statistical methods may influence stable gene identification [31,32]. Finally, the geometric mean value for the stability of each reference gene was used for a comprehensive analysis. Under different temperature conditions, the expression levels of *18S rRNA*, *PRS18*, and *TUB* (in order) were most stable in adults, and the expression levels of *TUB*, *18S rRNA*, and *SDHA* (in order) were most stable in larvae. These results not only provide an experimental basis for the selection of reference genes for studies of the cold tolerance of *D. valens*, but also provide a reference for screening studies focused on other insects. The application of these loci as reference genes under other physiological or experimental conditions remains to be determined.

## 5. Conclusions

In this study, six candidate reference genes were screened from transcriptome data for *D. valens*. The expression stability of the candidate reference genes in adults and mature larvae under temperature stress was evaluated. Based on our results, the combination of *18S rRNA* and *PRS18* is recommended for studies of gene expression in adult *D. valens* at different temperatures, and the combination of *18S rRNA* and *TUB* is effective for studies of gene expression in mature larvae. These results contribute to studies of reference genes in Coleoptera and provides a basis for molecular studies of *D. valens*.

## Figures and Tables

**Figure 1 insects-11-00328-f001:**
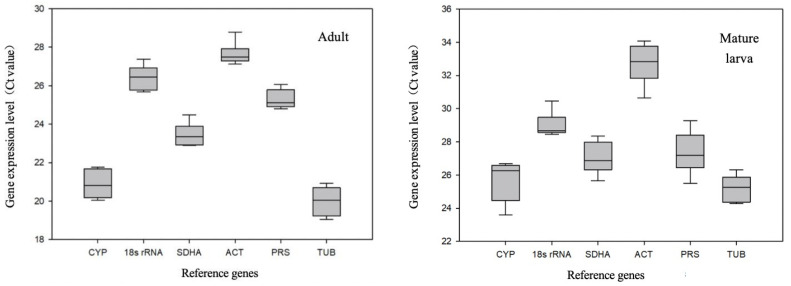
Expression levels of candidate reference genes in adults and mature larvae of *D. valens.* Expression levels are displayed as raw cycle threshold (Ct) values of the candidate reference genes of *D. valens* at different temperatures. The boxes represent the values between the 25th and 75th percentiles, the lines in the boxes indicate the median values, and the whisker caps denote the minimal to maximal values.

**Figure 2 insects-11-00328-f002:**
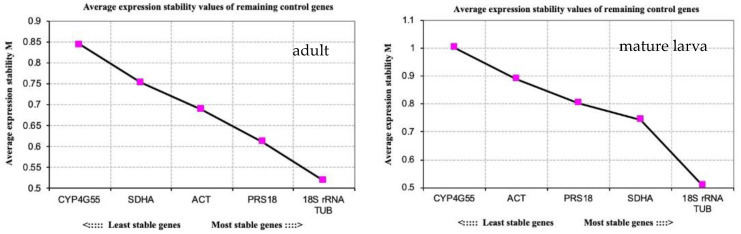
Expression stability of the candidate reference genes in adults and mature larvae of *D. valens* evaluated using geNorm.

**Figure 3 insects-11-00328-f003:**
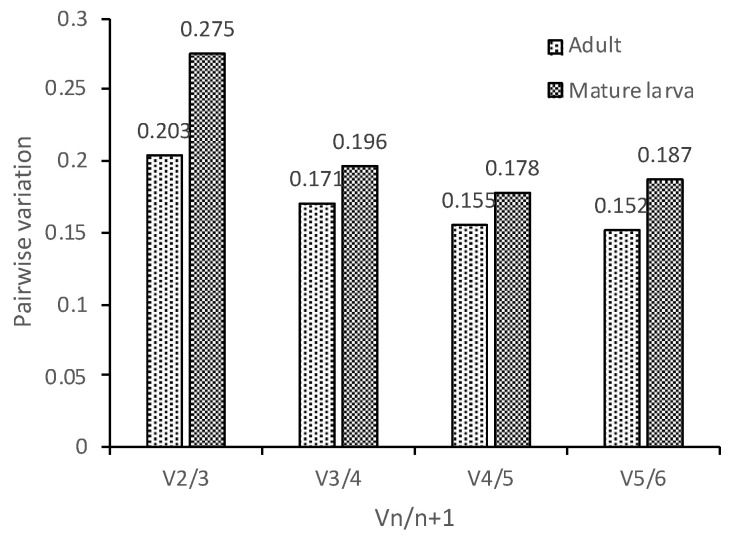
Pairwise variation (Vn/Vn + 1) analysis of the number of candidate reference genes in *D. valens.* According to the paired difference analysis of the standardized factor of the internal reference gene, the mutation value V was obtained for pairwise comparison, so as to determine the optimal number of the internal reference gene. When the Vn/n + 1 value was less than the threshold value of 0.15, the optimal number of the internal reference gene was n.

**Figure 4 insects-11-00328-f004:**
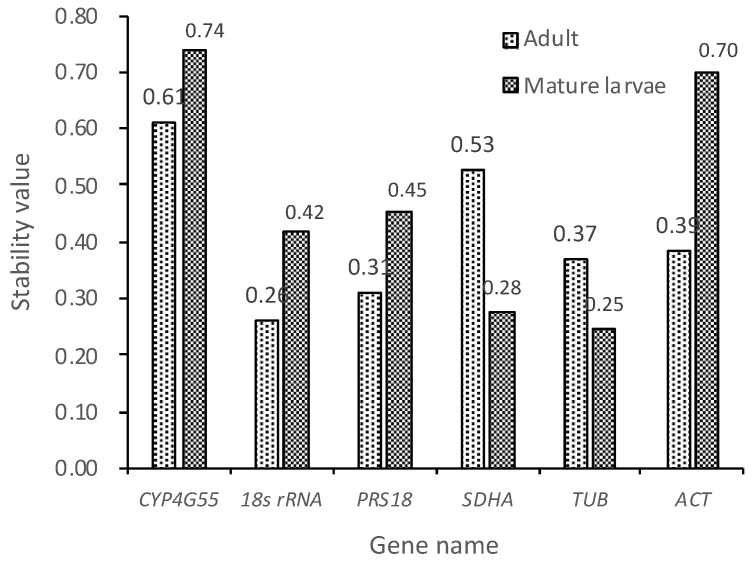
Expression stability of candidate reference genes in adults and mature larvae of *D. valens* evaluated using NormFinder. A lower stability value indicates a more stable expression.

**Figure 5 insects-11-00328-f005:**
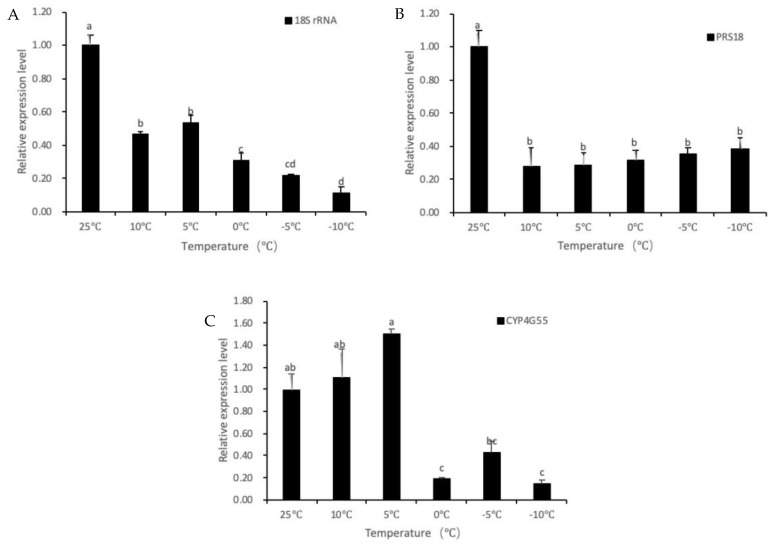
Relative expression levels of *HSP21* under various temperatures in adult *D. valens* using different reference genes. (**A**–**C**) *18S rRNA*, *RPS18* and *CYP4G55* were used as reference genes, respectively. Data are mean ± SE. Different letters above bars mean significant differences (*p* < 0.05, Duncan’s test).

**Table 1 insects-11-00328-t001:** Primer sequences, product sizes, and PCR efficiencies for the assessed genes.

Gene	RT-qPCR		
	Primer Sequences	Amplification Efficiency	*R* ^2^
*CYP4G55*	F: AGCCAACGAGTTCGGAAGAG	98.7	0.994
	R: TCAGAAGACCATCGCCCAAC		
*18S rRNA*	F: TGGAGGAAAACGGGCACTAC	91.3	0.998
	R: GACTTGTCTGCGTTGCACAG		
*SDHA*	F: CTGGTGCCGATACGCAAATG	92.1	1.000
	R: TAAGGCTGTTGTCGGACACC		
*TUB*	F: CTTACCACCCCCACATACGG	97.8	1.000
	R: ATTGCTGACTGCCTCTGGAC		
*ACT*	F: TGCTGCAGGAAGATCCACTG	109.8	0.995
	R: GCACTGTCCCTGTCAGGTAC		
*PRS18*	F: CATCGCTCTGTCCTCGGTAC	102.3	0.999
	R: TCGGTGTGCTTGACATCCAA		
*HSP21*	F: TGGATGTGGAGGGCTTCAAG	107.1	0.997
	R: TGTTAGAACGCCGTCCTCAC		

**Table 2 insects-11-00328-t002:** Expression stability of the candidate reference genes in adult *D. valens* evaluated using BestKeeper.

Gene	n	geo Mean	AR Mean	min	max	SD	CV [%]	min [x-fold]	max [x-fold]	SD [±x-fold]	CC [r]	*p*-Value
*CYP4G55*	18	20.83	20.88	19.22	24.21	1.20	5.77	−3.06	10.35	2.31	0.765	0.001
*18S rRNA*	18	26.39	26.42	24.07	28.70	1.01	3.81	−5.00	4.98	2.01	0.915	0.001
*SDHA*	18	23.42	23.45	21.52	26.15	0.87	3.72	−3.73	6.64	1.83	0.761	0.001
*ACT*	18	27.62	27.65	25.99	30.94	1.17	4.22	−3.09	10.00	2.24	0.878	0.001
*PRS18*	18	25.26	25.29	23.14	27.16	0.88	3.50	−4.35	3.73	1.85	0.876	0.001
*TUB*	18	19.90	19.99	16.40	23.09	1.65	8.26	−11.32	9.14	3.14	0.908	0.001

**Table 3 insects-11-00328-t003:** Expression stability of the candidate reference genes in mature larval *D. valens* evaluated using BestKeeper.

Gene	n	geo Mean	AR Mean	min	max	SD	CV [%]	Min [x-fold]	max [x-fold]	SD [±x-fold]	CC [r]	*p*-Value
*CYP4G55*	18	25.63	25.68	22.93	28.27	1.32	5.15	−6.50	6.26	2.50	0.773	0.001
*18S rRNA*	18	28.96	29.00	26.88	32.40	1.16	3.98	−4.24	10.82	2.23	0.836	0.001
*SDHA*	18	26.95	27.02	24.53	32.14	1.53	5.67	−5.35	36.53	2.89	0.905	0.001
*ACT*	18	32.66	32.71	29.05	35.18	1.62	4.95	−12.23	5.71	3.07	0.696	0.001
*PRS18*	18	27.28	27.33	24.44	32.01	1.36	4.97	−7.13	26.67	2.56	0.946	0.001
*TUB*	18	25.17	25.20	23.51	28.25	1.01	4.01	−3.17	8.46	2.01	0.931	0.001

**Table 4 insects-11-00328-t004:** Reference genes ranks by geNorm, NormFinder, BestKeeper, and overall rank.

	RANK	geNorm	NormFinder	BestKeeper	OVERALL
Adults	1	*18S rRNA/TUB*	*18S rRNA*	*SDHA*	*18S rRNA*
	2		*PRS18*	*PRS18*	*PRS18*
	3	*PRS18*	*TUB*	*18S rRNA*	*TUB*
	4	*ACT*	*ACT*	*ACT*	*SDHA*
	5	*SDHA*	*SDHA*	*CYP4G55*	*ACT*
	6	*CYP4G55*	*CYP4G55*	*TUB*	*CYP4G55*
Mature larvae	1	*18S rRNA /TUB*	*TUB*	*TUB*	*TUB*
	2		*SDHA*	*18S rRNA*	*18S rRNA*
	3	*SDHA*	*18S rRNA*	*CYP4G55*	*SDHA*
	4	*PRS18*	*PRS18*	*PRS18*	*PRS18*
	5	*ACT*	*ACT*	*SDHA*	*CYP4G55*
	6	*CYP4G55*	*CYP4G55*	*ACT*	*ACT*

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
