# Peer review of "Reference Gene Selection for Expression Analyses by qRT-PCR in Dendroctonus valens"

_insects, 2020, doi:10.3390/insects11060328_

Round 1

Reviewer 1 Report

The current manuscript is re-submission of earlier manuscript, authors have improved the manuscript still I have some concerns before accepting this manuscript.

Figure 2: the value in genorm studies should be represented at different temperatures if it is the average value than it should be explained in results section as well as in figure legends with statistical analysis.

Table 4: Rank 4: Larvae; please explain why SDHA is ranked when ACT was found in all three analyses, similarly why SDHA was ranked in rank 5. Same condition in adults rank 4 and 5 should be explained. Also justify explanation in results section 3.3.

Still there are some typos authors are recommended to read the manuscript carefully.

Reviewer 2 Report

In present study, the author analyzed the suitability of six housekeeping genes of D. valens as reference genes for real-time analysis under varied temperature conditions. The author used different software to evaluate expression stability of these genes, which is further verified by determining the relative expression level of HSP21 in adults. The study is well planned and executed but lacks novelty and has very limited application. Author can improve the manuscript quality by performing additional experiments that can enhance the applicability and readership of the study. The article should be checked by a native speaker, the grammar is sometimes not correct and articles/punctuations are missing quite often.

The specific comments are as follows:

-In the introduction, author should state, what is the rationale behind differential gene expression analysis under various temperature conditions in context to the effect of weather conditions on D. valens survival.

-Line 55-56, All the temperature conditions used in the study (-10°C, -5°C, 0°C, 5°C, and 10°C) are practically cold stress when compared to physiological temperature (25°C, used in this study). Author should include at least 37°C and 40°C temperature points in analysis to prove that reference genes are also stable during heat stress conditions. In addition, more time intervals to validate the stability of these reference genes during early and delayed temperature responses should be included in the experiment.

- Line 69-70, Author should provide a logical explanation for selecting these six genes only, why not other housekeeping genes? If it is previously reported then provide an appropriate reference.

- Single gene (HSP21) is not sufficient for data validation, author should use at least five previously reported genes that are differentially expressed during temperature stress.

- The author should be consistent with the writing, there many errors throughout the manuscript for example “HSP21” is used in the text while ”HSP20-1” is used in the figure legend.

-Primer sequence of HSP21 is missing in table 1.     

-Table 2, the value of n (18) is not clear, author should provide details of statistical analysis used in study.    

-Conclusion should be more precise comparing the current findings with previous reports.

Round 2

Reviewer 2 Report

I Agree to Accept in the present form.

This manuscript is a resubmission of an earlier submission. The following is a list of the peer review reports and author responses from that submission.

Round 1

Reviewer 1 Report

The manuscript provides evidence for choosing stable reference genes for qPCR analysis in the coleopteran pest Dendroctonus valens. The rationale of selecting the candidate reference genes is valid and described well. The reference genes are tested for stability using the normal algorithms and conclusions are drawn to provide a recommendation for qPCR studies on this insect with the usual caveats.

The introduction is well written, and the methods are written in a manner that would allow replication of the work by a competent scientist. Results for HSP21 expression demonstrate why a stable reference gene is vital.

Major concerns

There is a major disconnect within manuscript surrounding the role of temperature in the project. The temperature profiles are described in the methods yet are only considered at the end of the results section when three reference genes are used to define expression levels of HSP21. This is good and the results are clear. However, throughout the whole manuscript there are many claims that the selected reference genes are stable across the full range of temperatures. There is no evidence for this. Therefore, the conclusions drawn are not substantiated by the data present. Furthermore, there is confusion between result that are gained from the two different life stages and temperature. These need to be decoupled to provide full understanding and agreement with the findings. You must make it clear where this work on the stability of the reference gene over the full temperature range is presented in the manuscript and you must make it clear if a result refers to life stage or to the temperature of the insect.

Minor comments/corrections

Abstract

Lines 16 to 18 – expression levels are stated as stable, yet they are ranked for each life stage. This is confusing.

Line 19 – it would be more logical to reverse the order of 18s rRNA and TUB so that it matches the rank order given previously.

Introduction

Line 49 would benefit from a clear description of the aim and objectives the project rather than this brief description of the work undertaken.

Materials and Methods

Line 61 – some confusion, as it is written it suggests that the RNA extractions were performed at different temperatures, not that the RNA was extracted from the insects which had been stored at different temperatures.

Line 64 – include the amount of RNA used to synthesize the cDNA.

Line 69 – “assessed” should be “selected”

Line 70 – Following the RT-PCR step were these amplicons verified by Sanger sequencing? This is implied but not stated conclusively.

2.4 Primer design – please state the parameters used for the Primer3Plus design strategy. This will provide greater understanding for your readers.

Results

Line 113 – I do not see how figure 1 relates to different temperatures. Also, it would be easy to visualise the difference in transcript abundance if both panels in figure 1 used the same scale for the y-axis.

Lines 121, 122 and throughout – be consistent with the name of the analysis. Original authors defined the algorithm as geNorm.

3.2.2 – again clarification is needed to make the distinction between temperature and life stage.

Figure 3 – not referred to in the text. The pair-wise comparisons need full explanation, define V.

3.4 – define which temperature is the control. I can be assumed it is the 25°C group but needs clarification.

Line 173 – “not significantly lower…”. Why is this result given as a double-negative?. In addition to the observed contrast with 18s rRNA the result with CYP4G55 also contrasts with the data when PRS18 is used as the normaliser.

Discussion

Line 181 and 182 – This conclusion cannot be drawn from the data reported. At no point are the stability of the candidate genes analysed with respect to a range of temperatures.

Line 203 – this statement directly contradicts the abstract.

Line 209 – “reference” not “refer”.

Line 212 and 218 – clarify the role of temperature in these statements.

Figure and table legends

The legends need improving, they contain a title for the work but do not contain the necessary information to provide standalone interpretation of the data.

Figures 3 and 4 - These two figures need the x- and y-axis drawn.

Where a figure has multiple panels, you should add letters (a, b etc…) to allow the reader to easily identify the correct chart from the text.

Figure 5 – what is the purpose of the letters above the bars? These are not described in the caption, nor are they directly referred to in the text.

Reviewer 2 Report

In the Present manuscript “Reference gene selection for expression analyses by qRT-PCR in Dendroctonus valens” by Zheng et al, authors have explained reference gene selection for expression analysis in Dendroctonus valens. Six genes (ACT, TUB, SHDA, PRS18, 18S rRNA, and CYP4G55) were selected from transcriptome data of Dendroctonus valens. The 18S rRNA and PRS18 are recommended in combination for studies of gene expression in adults and the combination of 18S rRNA and TUB for mature larvae.  This work is extremely useful in light of expression analyses by qRT-PCR of Dendroctonus valens. I have some concerns before accepting this manuscript for publication.

  1. Authors are strongly recommended to rewrite whole manuscript carefully as English is not up to mark for publication.

  1. The entire work lack of depth and focus on the subject. The expression analysis of all the genes at different temperatures not shown in any experiment. Authors are recommended to elaborate all the experiments with more details.

  1. References are also not in same format, please correct them too.

Round 2

Reviewer 1 Report

Although the basic corrections have been changed in the revised manuscript the major issues; namely Points 9, 15, 16 and 19 have not adequately address. Overall there is still confusions as to what, with respect to the stability across the different temperatures, the manuscript is demonstrating.